# Self-Supervised Bug Detection and Repair

**Miltiadis Allamanis, Henry Jackson-Flux,**[*] **Marc Brockschmidt**
Microsoft Research, Cambridge, UK
{miallama, mabrocks}@microsoft.com

## Abstract

Machine learning-based program analyses have recently shown the promise of integrating formal and probabilistic reasoning towards aiding software development. However, in the absence of large annotated corpora, training these analyses is challenging. Towards addressing this, we present BUGLAB, an approach for self-supervised learning of bug detection and repair. BUGLAB co-trains two models: (1) a detector model that learns to detect and repair bugs in code, (2) a selector model that learns to create buggy code for the detector to use as training data. A Python implementation of BUGLAB improves by up to 30% upon baseline methods on a test dataset of 2374 real-life bugs and finds 19 previously unknown bugs in open-source software.

## 1 Introduction

Detecting and repairing bugs in source code requires strong reasoning skills over formal structures (*e.g.* data and control flow) and ambiguous information (*e.g.* identifier names, coding idioms, and comments). Traditional program analyses are able to detect critical bugs through formal reasoning and combinatorial search, but need to be manually coded by experts. That is a lengthy and costly process, which misses the opportunity to use ambiguous information pervasive within code.

Towards broadening the applicability of such methods, and utilizing ambiguous information, deep learning-based bug detection methods are being investigated [22, 3, 13]. These methods have the potential to further improve the engineering of software we rely on every day. However, many challenges in the area remain open, such as creating robust bug detection and repair methods that cover a wide range of common bugs in the absence of large supervised training corpora. Existing work focuses on randomly inserted bugs [22, 13], Cloze test proxy tasks [3], corpora of small code edits that *may* contain bugs [9] or build errors [28]. All these approaches rely on datasets of very limited size or ones known not to be representative of the characteristics of bugs found in real code.

In this work, we propose BUGLAB, a self-supervised approach that trains robust bug detectors by co-training a bug selector that learns to create hard-to-detect bugs (Sec. 2). For example, for a given code snippet with two well-named variables, a variable misuse bug may be easy to detect and repair, whereas an incorrect comparison operator might be significantly harder to identify. We propose a neural architecture for BUGLAB (Sec. 3) and implement it for Python (Sec. 4). Our implementation considers four broad classes of seemingly simple, yet hard-to-detect bugs and shows improved performance over training with randomly-inserted bugs on PYPIBUGS, a new, manually curated test set of 2374 real-life bugs (Sec. 5). Furthermore, we tested our trained models on popular open-source Python packages and identified 19 previously unreported bugs, though false positive rates of $\sim 98\%$ remain impractical. We hope that creating machine learning methods that can detect these bugs early and assist developers will speed up software development and allow engineers to deliver more robust software. We release PyPIBugs and our code at `https://github.com/microsoft/neurips21-self-supervised-bug-detection-and-repair`.

---

[*]Work done while at Microsoft Research.

35th Conference on Neural Information Processing Systems (NeurIPS 2021).

## 2   Self-Supervised Bug Detection

In this section, we first introduce the concept of code rewriting, and then use it to define BUGLAB as a framework for self-supervised learning of bug detection and repair.

**Code Rewriting**   Rewriting is common within compilers and their optimizations, test-driven search-based bug repair tools, mutation testing, and refactoring tools. Rewrites can be semantics-preserving (*e.g.* renamings of local variables), or semantics-altering (*e.g.* replacing `>=` by `!=`).

Let $\mathcal{S}$ denote the set of all syntax trees (not necessarily rooted in the start symbol of the language grammar). Syntax tree locations $\ell \in \{\epsilon\} \cup \mathbb{N}^*$ in a syntax tree $s \in \mathcal{S}$ are recursively defined, where $s_{|\epsilon} = s$ and $s_{|\ell}$ for $\ell = \ell' \circ i$ is the $i$-th child of $s_{|\ell'}$ (*i.e.* $s_{|(2,3)}$ denotes the third child of the second child of s). We define a rewrite rule $\rho = (m_\rho, t_\rho)$ as a pair of a matching function $m_\rho : \mathcal{S} \to \{true, false\}$ and a transformation function $t_\rho : \mathcal{S} \to \mathcal{S}$. The matching function $m_\rho(s)$ yields $true$ iff the rule $\rho$ is applicable at the root of a subtree s. The transformation function can be applied to obtain a transformed syntax tree. For convenience, we define $t_\rho(s) = s$ iff $m_\rho(s) = false$. We then write $\rho(s)$ to indicate the modification of a syntax tree s using $\rho$ when possible, and otherwise the identity function. For reversible rewrite rules $\rho$, we denote the inverse rule as $\rho^{-1}$ such that $\rho^{-1}(\rho(s)) = s$ holds. We discuss concrete rewrite rules $\rho$ in Sec. 4.

Given a set of rewrite rules $\mathcal{R}$ we define the set of "*potential rewrites*" in a syntax tree s as $R_s^{\mathcal{R}} = \left\{ \langle \ell, \rho \rangle \mid \rho \in \mathcal{R}, \ell \text{ location in } s, m_\rho(s_{|\ell}) = true \right\}$. For each tuple $\langle \ell, \rho \rangle \in R_s^{\mathcal{R}}$, we use $s' = s[\rho]_\ell$ to denote the new syntax tree obtained by applying $\rho$ at location $\ell$ of s. In BUGLAB, we train models that use rewrites from $R_s^{\mathcal{R}}$ to insert and repair bugs. We will discuss such neural models in Sec. 3.

**BUGLAB**   In BUGLAB, we are interested in self-supervised training of a robust *bug detector* model $D_\theta$ with parameters $\theta$ on an unannotated codebase $C$. Let $\mathcal{R}$ be a set of rewrite rules[2] that allows to insert and repair bugs. We train $D_\theta$ to be able to recognize the "hardest" possible rewrites that could be applied on our codebase $C$. For this, we consider the loss $\mathcal{L}_{D_\theta}$ of $D_\theta$ on a rewritten code snippet $s[\rho]_\ell$, for which the model needs to predict the repairing rewrite $\langle \ell, \rho^{-1} \rangle$. Formally, we want to minimize the objective

$$E_{s \sim C} \left[ \max_{\langle \ell, \rho \rangle \in R_s^{\mathcal{R}}} \mathcal{L}_{D_\theta} \left( s[\rho]_\ell, \langle \ell, \rho^{-1} \rangle \right) \right].$$

However, for any useful detector the set of rewrites $R_s^{\mathcal{R}}$ is commonly very large or unbounded and computing the maximum over all $\langle \ell, \rho \rangle \in R_s^{\mathcal{R}}$ is practically intractable. To address this, BUGLAB introduces a *bug selector* model $S_\phi$ (with parameters $\phi$), whose goal is to approximate the intractable $\max_{\langle \ell, \rho \rangle \in R_s^{\mathcal{R}}} \mathcal{L}_{D_\theta}(\cdot)$. We can then sample rewrites from $S_\phi$ instead of computing the maximum. We denote this as $\langle \ell, \rho \rangle \sim S_\phi(s)$ and the overall BUGLAB training objective can be written as a min-max optimization problem:

$$\max_{\phi} \min_{\theta} E_{s \sim C} \left[ E_{\langle \ell, \rho \rangle \sim S_\phi(s)} \left[ \mathcal{L}_{D_\theta} \left( s[\rho]_\ell, \langle \ell, \rho^{-1} \rangle \right) \right] \right]. \tag{1}$$

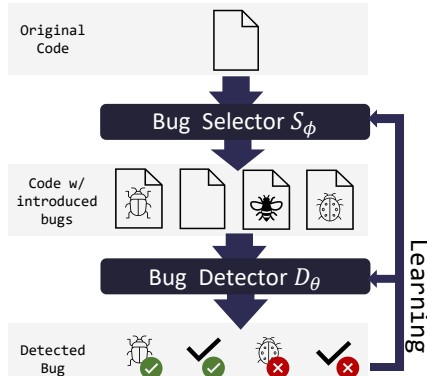

Figure 1: BUGLAB overview: a selector model $S_\phi$ decides which (bug-introducing) rewrite to apply to an input code snippet. Then a bug detector $D_\theta$ tries to locate and repair the inserted bug (if one was inserted).

The two models $S$ and $D$ in BUGLAB are "symmetric" in the sense that they both predict rewrites on code snippets, and only differ in their objectives — one aiming to introduce bugs and one aiming to repair them. In practice, we can and do use the same architecture to model both $S$ and $D$, which we will discuss in the next section. At test time, we discard $S$ and only use the trained detector $D$ to locate and repair bugs.

---

[2]In this work we assume that $\mathcal{R}$ contains a special "identity" rewrite rule $\rho_\varnothing$ that does *not* change the code.

# 3 Neural Models

In this section, we discuss how we represent code in BUGLAB and the neural models we use to learn how to rewrite code in the selector and detector models.

**Code Representation** We consider source code as a set of entities $v_i \in V$ which relate to each other with a set of typed relations $e_k \in E$, where a relation $e_k = (v_i, r, v_j)$ denotes a relationship between entities $v_i$ and $v_j$ with type $r$. The entities and relations can be thought as a heterogeneous graph $G = (V, E)$. The choice of code entities and their relationships is a form of high-level feature extraction. We discuss concrete entities and relationships for Python in Sec. 4. We also define a projection function $\mathbb{P}_{tok}$ that accepts $V$ and $E$ and returns a *sequence* $V_{tok}$ of the token entities in $V$ with the nodes appearing in relations in $E$ deterministically mapped to elements of $V_{tok}$, *i.e.* $E_{tok} = \{(p(v_i), r, p(v_j))\}$, where $p$ maps the entities in $V$ to $V_{tok}$. $\mathbb{P}_{tok}$ will be used for relational transformer models.

To learn a neural representation of the code entities $v_i$, first we define an embedding function $\boldsymbol{e}(v_i)$ which maps the content of each entity to an initial $D$-dimensional representation. Throughout this work — similar to Allamanis et al. [4] and other previous work — we deterministically split the string representation of each node into subtokens (*e.g.*, `fooBar` is split into `foo` and `bar`), embed them through a learned embedding matrix, and use max pooling to get a single vector. We then "contextualize" the entity representations within $G$ using one of two models: a MLP-based GNN model with max message aggregation and the GREAT relational transformer of Hellendoorn et al. [13] over the token sequence and relations $V_{tok}, E_{tok} = \mathbb{P}_{tok}(V, E)$. GREAT uses both positional encodings and the projected relations in $E_{tok}$. See Appx. A for detailed architecture descriptions. Other models to compute entity representations can be used, but were not explored in this work.

We use $\boldsymbol{r}_\ell$ to denote the computed vector representation of the entity at location $\ell$, independent of the model used to produce it. We use these representations to define our code rewriting models.

**Probabilistic Code Rewriting Models** Both bug selection and bug detection require to model the probability of applying a specific rewrite at a location in a code snippet s, either to introduce or repair a bug. For this, we factorize this task into localization and rewrite-given-location models, *i.e.*

$$p\left(\langle \ell, \rho \rangle \mid \mathrm{s}, R_\mathrm{s}^\mathcal{R}\right) = p_{loc}\left(\ell \mid \mathrm{s}, R_\mathrm{s}^\mathcal{R}\right) p_{rew}\left(\rho \mid \ell, \mathrm{s}, R_\mathrm{s}^\mathcal{R}\right). \tag{2}$$

We model $p_{loc}$ as a probability distribution over the relevant locations $\{\ell \mid \langle \ell, \rho \rangle \in R_\mathrm{s}^\mathcal{R}\} \cup \{\texttt{NoBug}\}$, where `NoBug` is a special location used to indicate that the code is not buggy. In practice, we implement this similar to a pointer net [19] using the representations $\boldsymbol{r}_\ell$ (see Appx. A for details).

To select rewrites, we use rewrite type-specific learnable *rule score functions* $w_\rho\left(\boldsymbol{r}_\ell, \mathcal{M}_\rho(\mathrm{s}, \ell)\right)$. This function maps a vector representation of an entity $\boldsymbol{r}_\ell$ and potential additional metadata onto a scalar score. The rule-specific metadata $\mathcal{M}_\rho(\mathrm{s}, \ell)$ is defined for some rewrites, *e.g.* containing representations of other entities that could be used in the location $\ell$. We will discuss three concrete rule score functions in Sec. 4. The rewrite probability distribution $p_{rew}$ is then modeled by a softmax over the scores of all applicable rewrites at a target location $\ell$, *i.e.*

$$p_{rew}\left(\rho \mid \ell, \mathrm{s}, R_\mathrm{s}^\mathcal{R}\right) = \operatorname*{softmax}_{\langle \ell, \rho' \rangle \in R_\mathrm{s}^\mathcal{R}}\left(w_{\rho'}\left(\boldsymbol{r}_\ell, \mathcal{M}_{\rho'}(\mathrm{s}, \ell)\right)\right).$$

# 4 A Python Implementation

This section presents an implementation of BUGLAB for Python called PYBUGLAB. PYBUGLAB currently tackles a large subset of "stupid simple bugs" [16]. Fixing these bugs requires small changes to the code, but commonly has significant impact on code correctness. Such bugs may be thought as a form of a typographical mistake or a copy-paste error, and are often relatively hard to locate by humans but obvious after the fact. They are also quite common, as observed in the empirical statistics of Karampatsis and Sutton [16] and Just et al. [14]. Future work may focus on a broader set of rewrite rules or even learnable rewrites, but as we will observe in Sec. 5 more work is needed towards this. Almost all ideas in PYBUGLAB transfer straightforwardly to other programming languages other than Python, but would require some engineering effort to implement.

**PYBUGLAB Code Entities and Relations** In this work, we follow related literature (see Sec. 6 for more) and extract entities and relationships that are readily available by tokenizers, parsers,

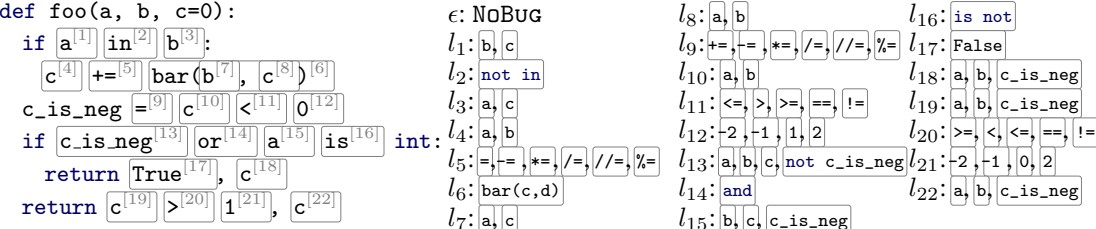

Figure 2: Code snippet and rewrites available to PYBUGLAB.

existing simple program analyses, or other Python-specific program analysis tools. The complete list of entities and relationships can be found in Appx. B and include syntactic entities and relations, relations about the intraprocedural data and control flow, types, and documentation. Some notable entities include SyntaxNodes, Tokens, and Symbols (references to variables and functions). Fig. 4 in Appx. B shows a graph of the entities and relationships of the snippet in Fig. 2.

### 4.1 Bug-Inducing PYBUGLAB Rewrite Rules

PYBUGLAB focuses on four common kinds of bugs. Fig. 2 shows a code snippet and the rewrites allowed for each location, which number 63 even for this small example.

**Variable Misuse** Originally defined by Allamanis et al. [3] as a Cloze test for source code, Vasic et al. [30] and Hellendoorn et al. [13] reformulated the task to localizing a variable misuse bug (if any) within a snippet and repairing it. PYBUGLAB uses the latter representation. Variable misuse bugs are common, with 12.8-14.8% found in the ManySStuBs4J corpus [16] and about 6% of them caught during Java compilation in the Google build system [28]. To insert and repair variable misuse bugs, PYBUGLAB supports variable-swapping rewrites, such as in locations $l_1$, $l_3$ and $l_4$ (amongst others) in Fig. 2. To score a variable-swapping rewrite, we use the representation of the rewrite location $r_\ell$ along with the representation $r_\sigma$ of a variable Symbol $\sigma$ that could replace the current variable, *i.e.* is in-scope and has been defined before $\ell$. The rule score function $w_\rho$ for replacing the variable at $\ell$ with the symbol $\sigma$ is then computed as the inner product $r_\ell^\top r_\sigma$.

**Argument Swapping** (or Argument Selection) First coined by Rice et al. [26], it refers to swapping the arguments of a function invocation, *e.g.* in $l_6$ of Fig. 2. Rice et al. [26] and DeepBugs [22] tackled this problem when all arguments are single identifiers. PYBUGLAB extends this to swapping arbitrary argument expressions. The rule score function $w_\rho$ for an argument swapping rewrite is a two-layer MLP applied to the concatenation of the output representations of the representation of the parameter and the to-be-swapped arguments `arg1`, and `arg2`: $MLP\left([r_{\text{params}}, r_{\text{arg1}}, r_{\text{arg2}}]\right)$.

**Wrong Operator** Corrupting operators has a long history in mutation testing [14]. Detecting incorrect operators with deep learning was first tackled by DeepBugs [22] by using learnable embeddings of operators, operands and literals for arithmetic and comparison operators. DeepBugs focused only on binary operators. In PYBUGLAB we tackle all binary operators, including Boolean, arithmetic and comparison operators and two unary operators: logical and arithmetic negation. Locations $l_{11}$, $l_{14}$, $l_{16}$, and $l_{20}$ in Fig. 2 are rewrites related to wrong operators. The rule score function $w_\rho$ for an operator rewrite again uses an inner product, $r_\ell^\top r_{\text{op}}$, where $r_{\text{op}}$ is a learned embedding for operator op. Note that we rewrite operators only to compatible operators (*e.g.* < to > but not +).

**Wrong Literal** Corrupting operands, and specifically, literals appearing in the source code, is also a common strategy in mutation testing. As in mutation testing, PYBUGLAB handles a limited number of commonly used literals, allowing rewrites to replace integer literals within the set of -2, -1, 0, 1, 2 and swapping the Boolean literal `True` with `False` and vice versa. The scoring function is identical to the operator rewrite, using a learnable embedding $r_{\text{lit}}$ for each literal `lit`.

### 4.2 PYBUGLAB Rewrite Rules for Data Augmentation

We additionally consider more rewrite rules that are not meant to change the program semantics, using them as a form of data augmentation. This is in spirit similar to ideas in computer vision where images are transformed (*e.g.* rotated, cropped) but maintain their original content. Such rewrites

---

**Algorithm 1** Sequential Training Procedure for Selector and Detector models

---

**Require:** Code dataset $C$, initial detector/selector model parameters $\theta^{(0)}$, $\phi^{(0)}$

1: **for** meta-epoch $i = 0$ to $I$ **do**
2:     // Create dataset of buggy programs:
3:     $C_D^{(i)} \leftarrow \left\{ \left( \mathrm{s}[\rho]_\ell, \langle \ell, \rho^{-1} \rangle \right) \mid \mathrm{s} \in C, k \text{ samples } \langle \ell, \rho \rangle \sim S_{\phi^{(i)}}(\mathrm{s}) \right\}$
4:     $\theta^{(i+1)} \leftarrow$ update $\theta^{(i)}$ by training $D$ on $C_D^{(i)}$
5:     // Create dataset of hard-to-detect bugs:
6:     $C_S^{(i)} \leftarrow \left\{ \left( \mathrm{s}, \arg\max_{\langle \ell, \rho \rangle \in R_\mathrm{s}^{\mathcal{R}}} \left( \mathcal{L}_{D_{\theta^{(i+1)}}} \left( \mathrm{s}[\rho]_\ell, \langle \ell, \rho^{-1} \rangle \right) \right) \right) \mid \mathrm{s} \in C \right\}$
7:     $\phi^{(i+1)} \leftarrow$ update $\phi^{(i)}$ by training $S$ on $C_S^{(i)}$

---

have been shown to yield adversarially robust models of code [23]. Although our goal is *not* to provide adversarial robustness, we believe that such rewrites can help generalization. PYBUGLAB implements the following rewrites for this purpose:

- **Variable Renaming** renames a local variable to a random name not already in scope.

- **Comment Deletion** removes code comments, including docstrings and inline comments. Such comments commonly contain natural language information that is useful for code comprehension, but usually do not affect program semantics.

- **Comparison Expression Mirroring** swaps the two sides of a comparison operator and changes it appropriately. For example, `a<b` is transformed to `b>a`. Note that in cases such as `foo() < bar()`, this will change the order of execution of `foo` and `bar`, possibly altering program semantics.

- **If-Else Branch Swapping** negates the test condition of an `if-else` statement or a ternary expressions using DeMorgan's law and swaps the `then` body with the `else` body.

### 4.3 Implementation Details

To make the training computationally tractable we approximate Eq. 1. A simplified, sequential version of our training procedure is shown in Alg. 1. Intuitively, we alternate between training the two models, as the (discrete) sampling of rewrite rules in the selector models precludes direct end-to-end training. We first use the current state of the selector model to generate "hard" samples and train the detector model on these samples (we always include the unmodified (*i.e.*, NOBUG case) as a sample). Then, we use the loss of the detector model to identify those generated samples that were hardest to detect and train the selector model to produce such samples.

In practice, we implemented the training procedure as a system of asynchronously communicating processes, and all of the described steps happen in parallel. We do not use "generations" $C_{D/S}^{(0)}, C_{D/S}^{(1)}, \ldots$ of datasets, but instead use two constantly updated "pools" of training data, one for the detector and one for the selector. Each training step samples a minibatch from the current state of the corresponding data pool. We remove samples from the data pool once they have been sampled $\nu$ times for use in training, in spirit similar to replay buffers in reinforcement learning. In our experiments, $\nu$ was set to 4. We regularly (in separate, concurrent processes) take snapshots of the the current state of the $D$ and $S$ models to generate new elements that are updated to the data pools, matching the procedure described in Alg. 1. We approximate the $\arg\max$ in line 6 by only considering the $k$ samples chosen in line 3 for each input program. During training of $S$, we then mask out the unobserved choices before computing the loss.

## 5 Evaluation

We now discuss our new dataset and evaluate PYBUGLAB. We highlight key results.

**Datasets** To train PYBUGLAB we retrieve the 4k most downloaded packages in the Python package index (PyPI) and take 3.4k of them as training packages, using the rest for test purposes. During training, PYBUGLAB installs each package along with all its dependencies. Installing all the dependencies is important for extracting the entities and the relations beyond local syntactic ones (*e.g.* type

inference, method resolution). For each file, PYBUGLAB checks if it is a duplicate of a file that has already been seen in the training following the method of Allamanis [1] and runs all the relevant program analyses to extract the entities and relationships in each function. When we use additional rewrites for data augmentation, these are applied at the input of the PYBUGLAB pipeline as a form of pre-processing. Following Alg. 1, the bug selector $S$ selects $k = 5$ bugs to introduce, rewrites the source code text, and then the program analyses extract the new entities and relationships for the rewritten code snippets. The initial and rewritten code snippets are then used to create the training data for the detector and selector models.

We use two testsets to measure performance. First, we create **RANDOMBUGS**, a testset of 761 445 snippets derived from functions from the 600 PyPI test packages (not seen during training). For each function we find within these packages we add it to the dataset along with 9 rewritten functions with a randomly inserted bug. On average graphs have 260 nodes, 601 edges, 25 rewrite locations, and 130 possible rewrites. We also collect a testset of real bugs. Although we conjecture that, in practice, the vast majority of bugs like those discussed in Sec. 4.1 are fixed when developers locally test their software, a few of those slip and then are fixed across different revisions checked into a version control systems. We have crawled the accessible repositories of all 285k packages in the Python Package Index (PyPI), collected and manually filtered bugs captured by the rewrites from Sec. 4.1. This new dataset, **PYPIBUGS**, contains 2374 real-world, small bugs. We describe the data collection process in detail in Appx. D. In addition, we consider PYPIBUGS-PostFix: the examples from PYPIBUGS *after* a bug was fixed - we believe these samples are very likely to not contain any bugs anymore. We publish the dataset at `https://www.microsoft.com/en-us/download/103554` and include it in the supplementary material.

## 5.1 Quantitative Evaluation

Our first experiment aims to evaluate whether the BUGLAB training framework yields more precise bug detectors. We consider two model architectures, using either GNNs or the GREAT transformer to compute embeddings of code entities (architecture details and hyperparameter choices can be found in Appx. A). We use four different training strategies: "supervised" is training only a bug detector on a fixed dataset of 1 million functions from the 3.4k training packages with randomly inserted bugs. "Random Selector" refers to a variant of PYBUGLAB using a bug selector model that uniformly at random picks a rewrite to insert bugs. Finally, PYBUGLAB and PYBUGLAB +Aug use our framework from Sec. 2, with the latter also using additional rewrites to augment our code corpus. For the fully supervised model, we train with early stopping over a validation set; the other models are trained for a fixed number of 300 epochs (with 200k training samples per epoch) for the bug detector[3] and the last detector model is used for evaluation.

**Effectiveness of BUGLAB Training**
We first consider the performance of different models on the synthetic RANDOMBUGS dataset. Tbl. 1 shows the accuracy of predicting a full bug repair correctly ("Joint") and analogous to Eq. 2 break this up into a localization accuracy ("Loc") of predicting the correct location (or NOBUG for correct examples) and a repair accuracy ("Repair") for selecting the correct rewrite given the buggy location.

Table 1: Accuracies (%) for different training strategies and model architectures on RANDOMBUGS.

| | RANDOMBUGS | | | | | |
| | GNN | | | GREAT | | |
| | Joint | Loc | Repair | Joint | Loc | Repair |
| --- | --- | --- | --- | --- | --- | --- |
| Supervised | 62.4 | 73.6 | 81.2 | 51.0 | 61.9 | 76.3 |
| Random Selector | 69.4 | 79.6 | 84.0 | 63.9 | 73.6 | 82.0 |
| PYBUGLAB | 69.6 | 80.4 | 84.2 | 64.0 | 74.3 | 82.3 |
| PYBUGLAB +Aug | **70.3** | **81.1** | **84.5** | 65.3 | 75.3 | 82.5 |

We observe that PYBUGLAB-training leads to more robust bug detectors compared to other methods for both GNNs and GREAT. Random selector models — a form of data augmentation — improve performance over supervised methods but mostly on in-distribution RANDOMBUGS samples. As expected, augmenting the code dataset helps generalization, but does not make a substantial difference. Expanding the kinds of rewrites used to augment the data and learning to select them may improve performance in the future.

---

[3]This amounts to about 1.5 weeks for the GNN models and about 1 week for the GREAT models on a single P100 GPU.

Table 2: Results for different training strategies and model architectures on PYPIBUGS.

| | PYPIBUGS | | | | | | PYPIBUGS-PostFix | | | |
| | GNN | | | GREAT | | | GNN | | GREAT | |
| | Joint | Loc | Repair | Joint | Loc | Repair | Loc | Joint AUC | Loc | Joint AUC |
|---|---|---|---|---|---|---|---|---|---|---|
| Supervised | 20.0 | 28.4 | 61.8 | 16.8 | 25.8 | 58.6 | 17.8 | 0.087 | 20.7 | 0.044 |
| Random Selector | 21.2 | 27.0 | 69.2 | 20.6 | 26.8 | 67.2 | 47.5 | 0.108 | **52.5** | 0.117 |
| PYBUGLAB | 24.2 | 31.3 | 70.7 | 24.0 | 32.8 | 67.9 | 32.9 | 0.160 | 28.6 | 0.140 |
| PYBUGLAB +Aug | **26.4** | **33.5** | **72.0** | 23.2 | 29.7 | 68.8 | 32.6 | **0.187** | 48.2 | 0.129 |

Table 3: Localization and Repair Accuracy (%) per bug kind for the PYBUGLAB +Aug model.

| | RANDOMBUGS | | | | PYPIBUGS | | | |
| | GNN | | GREAT | | GNN | | GREAT | |
| Bug Type | Loc | Repair | Loc | Repair | Loc | Repair | Loc | Repair |
|---|---|---|---|---|---|---|---|---|
| Argument Swapping | **85.0** | **57.3** | 65.5 | 57.2 | **33.2** | **73.9** | 24.3 | 72.7 |
| Wrong Assign Op | **96.1** | **99.1** | 94.5 | 98.6 | **20.0** | **68.9** | 14.0 | 58.1 |
| Wrong Binary Op | **83.0** | **85.2** | 77.3 | 81.4 | 27.2 | **54.3** | **36.6** | 43.7 |
| Wrong Boolean Op | **71.8** | **99.5** | 43.6 | **99.5** | **27.6** | 96.9 | 15.7 | **97.2** |
| Wrong Comparison Op | **83.9** | **79.3** | 80.0 | 76.4 | **33.7** | **66.1** | 31.1 | 53.5 |
| Wrong Literal | **71.7** | **74.7** | 66.6 | 71.6 | **21.6** | 78.4 | 17.9 | **79.5** |
| Variable Misuse | **84.9** | **88.4** | 78.2 | 86.3 | **35.3** | **70.5** | 34.0 | 69.4 |
| NoBug | 53.8 | — | **62.5** | — | — | — | — | — |

Furthermore, bug localization is much harder than repair at a given location. This is somewhat expected: there are many more candidate locations compared to potential repairs at a given location. However, this suggests that research should focus on the localization problem rather than repair.

We now turn to the results on PYPIBUGS, shown in Tbl. 2, which also includes the accuracy of choosing the special NoBug location on the PYPIBUGS-PostFix dataset, as well as the area under the precision recall curve for the results on both PYPIBUGS and PYPIBUGS-PostFix.

We find that detecting and repairing real-life bugs is significantly harder than handling randomly inserted bugs. As PYBUGLAB models trained using a learned bug selector outperform those using a "Random Selector", we speculate that the learned selector avoids generating easy-to-detect bugs, focusing the detector model on recognizing deeper semantic patterns. Despite this, improvements in RANDOMBUGS often correlate with improvements in PYPIBUGS. This is encouraging: collecting PYPIBUGS-like datasets is costly; corpora with random bugs can help measure relative improvements to some extent. Finally, we find that recognizing non-buggy samples is very hard, and in particular, does not always profit from training in PYBUGLAB.

In our qualitative analysis (Sec. 5.2), we observed that the models raised some confident but incorrect warnings at very "odd" locations. However, these warnings were different across models. We have tested an ensembling strategy averaging the output probabilities of five separately trained GNN models. This results in localization and repair accuracies of 83.0% and 85.4% on RANDOMBUGS (*vs.* 81.1% and 84.5%) and 34.4% and 72.2% on PYPIBUGS (*vs.* 33.5% and 72.0%). As we discuss in Sec. 5.2 finding the cause of the "spurious" warnings is important future work.

**Per-Bug Evaluation** To better understand which bugs are hard to detect, we break down the results the best-performing PYBUGLAB +Aug models on RANDOMBUGS by type of bug in Tbl. 3. We observe that incorrect literals are some of the hardest bugs to detect. Incorrect assignment operators (*e.g.* = and +=) are easy to detect in RANDOMBUGS, but significantly harder in PYPIBUGS. This may be attributed to class imbalance, with simple assignment (=) being the majority class. Detecting if a snippet has a bug or not seems to be the hardest task: no model achieves accuracy beyond 63%.

We note that in our experiments, GNNs-based models seem to often outperform GREAT, somewhat contradicting the results of Hellendoorn et al. [13]. We have performed substantial additional experiments to investigate and verify these results, cf. Sec. A.2. This may have to do with the performance of these models on long sequences or that the GNN has access to more fine-grained information, instead of relations over the projected token sequences. For example, this could be attributed to the lack of syntax and symbol nodes in the representation used in GREAT. Nevertheless, GREAT

Table 5: Bug distribution (%) in different datasets

| Bug Kind | PYPIBUGS | RANDOMBUGS | Selector Samples |
|---|---|---|---|
| Argument Swapping | 11.9 | 8.4 | 23.8 |
| Wrong Assignment | 1.9 | 8.5 | 5.3 |
| Wrong Binary Operator | 3.4 | 2.4 | 2.3 |
| Wrong Boolean Operator | 8.1 | 2.2 | 6.4 |
| Wrong Comparison Operator | 17.1 | 8.2 | 7.4 |
| Wrong Literal | 3.7 | 11.6 | 12.4 |
| Variable Misuse | 53.8 | 58.6 | 42.5 |

is noticeably better (62.5% *vs.* 53.8%) at detecting NoBug and locating wrong binary operators in PYPIBUGS.

**Bug Selector Performance** To understand how training of the bug selector proceeds, we perform two experiments. In our first experiment, we take a snapshot of the selector model during training of the PYBUGLAB +Aug (GNN) model every 24 hours, after an initial burn-in phase of 12 hours. We then generate 10000 buggy samples using each of these snapshots and then test a fixed model on each of these snapshots. The results of this are shown in Tbl. 4, using a fully trained PYBUGLAB +Aug (GNN) model from another training run as a fixed model. We conclude that PYBUGLAB succeeds in learning to generate harder to find bugs, though we can observe the selector model trading off "harder-to-localize" and "harder-to-fix" properties. Tests on other models show similar trends, confirming the robustness of this result.

Table 4: Development of Performance on Bug Selector Samples

| Training Time | Joint | Loc | Repair |
|---|---|---|---|
| 0.5 days | 64.2 | 83.8 | 72.1 |
| 1.5 days | 62.5 | 80.7 | 72.9 |
| 2.5 days | 62.0 | 83.0 | 69.8 |
| 3.5 days | 61.7 | 82.5 | 69.8 |
| 4.5 days | 61.9 | 83.0 | 69.5 |
| 5.5 days | 61.1 | 83.0 | 68.6 |
| 6.5 days | 60.5 | 78.7 | 72.4 |

In a second experiment, we compare the distribution of different bug kinds in PYPIBUGS and RANDOMBUGS with the distribution of bugs sampled from the final snapshot of our selector model from above. The results are shown in Tbl. 5, where we can see that a number of bugs (argument swapping, use of wrong literals and of assignment operators) are substantially over-represented, whereas mistakes in comparison operators and variable misuse are under-represented. This indicates that PYBUGLAB generates hard to find, but not necessarily realistic bugs.

**Comparison to CuBERT** Finally, we compare our models to CuBERT [15], which uses a masked language modeling objective to pre-train a BERT-like model and then learns bug detectors specific to a class of bugs (*e.g.*, wrong binary operators) on top of this pre-trained model. Note that CuBERT detects *if* a bug exists but does *not* localize it. For the comparison, we create two sub-datasets of PYP-IBUGS: PYPIBUGS-WrongOp contains the 501 samples that involve the binary operators supported by CuBERT, and PYPIBUGS-VarMisuse, which contains the 1278 bugs that involve variable misuses. We complete both of these datasets with 501 (resp. 1278) random NoBug code samples from our RANDOMBUGS, to match the 1:1 buggy/non-buggy distribution used in CuBERT's training. Since CuBERT classification models focus on a single bug type, to compare to PYBUGLAB we mask out all code locations that do *not* correspond to a bug that could be detected by the corresponding CuBERT model. We then treat the prediction of the NoBug location as a "non-buggy" prediction and all other locations as a "buggy" prediction. For example, for the snippet in Fig. 2, only the locations $l_2$, $l_{11}$, $l_{14}$, $l_{16}$, and $l_{20}$ and their corresponding rewrites are considered by PYBUGLAB for the comparison on PYPIBUGS-WrongOp.

Tbl. 6 shows the results of comparing the released CuBERT snapshots with the PYBUGLAB +Aug GNN model. We observe that the PYBUGLAB models

Table 6: Comparison with CuBERT [15]

| | CuBERT | | | PYBUGLAB (GNN) | | |
|---|---|---|---|---|---|---|
| | Prec | Recall | F1 | Prec | Recall | F1 |
| PYPIBUGS-WrongOp | **0.764** | 0.251 | 0.378 | 0.730 | **0.764** | **0.746** |
| PYPIBUGS-VarMisuse | 0.632 | 0.403 | 0.493 | **0.740** | **0.840** | **0.787** |

have substantially better recall than CuBERT-based models, even though they were trained to detect more bug types. When calibrating the CuBERT models to have a recall equal to PYBUGLAB, their precision drops substantially. In particular, on PYPIBUGS-WrongOp, it is reduced to 0.609, and

```
1 def make_id(name):
2   r = get_rand_string(12)
3   if len(name) <= 22:
4       name = name[:22]
5   return name + "-" + r
```

(a) A wrong comparison operator bug (red box) in PYPIBUGS detected and repaired by the GNN PYBUGLAB +Aug models.

```
1 def update(self, roomId,
2           title, **request_params):
3   check_type(roomId, basestring)
4   check_type(roomId, basestring)
5   [...]
```

(b) A variable misuse (red box) caught in an open-source project. GNN PYBUGLAB +Aug suggests to rewrite roomId to title. The fixing pull request is found here.

Figure 3: Bugs found by PYBUGLAB. Snippets reformatted and abbreviated to fit figure.

on PYPIBUGS-VarMisuse, it is reduced to $0.613$; in both cases, PYBUGLAB outperforms CuBERT substantially.

## 5.2  Qualitative Inspection of Raised Warnings

We now take a qualitative look at the raised warnings raised by PYBUGLAB. As example, Fig. 3a shows a sample of PYPIBUGS where the developer used an incorrect comparison operator. Once pointed to it, it is clear to a human that the truncation statement in line 4 has no effect (under the reasonable assumption that `name` is a string), and that a different comparison operator (`>`) is necessary.

To gain an understanding of the performance of PYBUGLAB on realistic data, we performed an in-depth analysis of the cases flagged as bugs by our best-performing model on the code found within the 4k top PyPI packages. We observed a mixture of false positives with few previously unseen real-life bugs, matching the quantitative results in Tbl. 3. First, we find that the majority of the false positives are "incorrect literal" detections. This suggests that learning to detect such bugs is a hard problem. Furthermore, many literals serve as default "configurations" (*e.g.* the number of retries for a network request) and different values are *not* bugs. We posit that a large percentage of literal replacements the selector learns to make fall in this category.

We also found that some repairs suggested by the model actually produce semantically equivalent code. For example, the model lacks knowledge that two variables refer to the same object in memory (aliasing), and so attempts to "repair" variable misuse bugs by switching between these. Other examples includes checking the return values of standard functions such as Python's `str.find`, which returns $-1$ if the query string is not found. In such cases, PYBUGLAB often suggested to rewrite an `if x.find(y) <= -1` to `if x.find(y) == -1`, which makes no difference in practice. These false negatives can be attributed to the fact that the bug selector model considers such changes as introducing bugs, even though they are not actually changing behavior. This suggests that for better results, the rewrite rules need to ensure that the rewrites are *not* semantics-preserving and represent bugs.

Finally, some reported issues were sufficiently complex that it took us (the human authors) a couple of minutes of thought to conclude that a warning is spurious. Simultaneously, there are some warnings that are "obviously" incorrect to us, but the reasons why the neural models raise them is unclear. This highlights the importance of research on explainability techniques along with better ways to calibrate model confidence. The fact that selectors may introduce spurious "bugs" may also be affecting how the detector model learns. Ideas that have appeared in reinforcement learning, such as the one of Dennis et al. [8], may allow models to improve their performance in spite of spurious bugs.

Overall, only 19 of the 1000 reported warnings were found to be real-life bugs. Of these 19, we reported 11 on GitHub (6 already merged, 5 pending approval). See Appx. G for details. 3 other bugs had already been fixed between the version PYBUGLAB processed and the current version or the project was deprecated, whereas another 5 bugs are minor and we decided not to report them. One of the detected bugs is shown in Fig. 3b. Overall, most of the detected bugs appear within unit tests, logging, or exception handling, possibly because bugs there do not impact the core functionality of a project. However, given the number of such bugs we collected in PYPIBUGS, we believe that such bugs arise equally often in other code, but that they are detected and fixed more quickly.

Although our analysis only forms a lower bound on the precision of PYBUGLAB and related methods, it suggests that there is still ample room for future improvements towards making machine learning-based bug detection and repair practically useful.

## 6 Related Work

Detecting bugs in source code has been researched since the early days of computing. Traditionally, bug detection is tackled as a formal task, where any code that cannot be proved to satisfy some correctness property may contain a bug. This is essential for security- and safety-critical bugs, but not for other — equally common — bugs. In the last decade, software engineering and programming language research have increasingly realized ambiguous information within code (*e.g.* variable names, comments) contains valuable information and using this information can yield valuable results [2]. The main premise is that patterns in source code, such as patterns in names, control, and data flow can be informative. This information can also be exploited to detect some bugs. For example, Ray et al. [24] noted that even simple language models tend to assign lower probability to buggy code.

Multiple static analysis methods have been researched that combine some form of data-oriented bug detection. This ranges from language model-based tools, such as the early work of Wang et al. [32] to specification-mining tools such as the work of Eberhardt et al. [10]. BUGLAB is related to DeepBugs [22] which uses an MLP over a limited window of code tokens and train separate models to detect wrong operators, operands, and argument swappings. BUGLAB opts for a more structured representation of code and a single model. Allamanis et al. [3], Vasic et al. [30], Hellendoorn et al. [13] tackle variable misuse bugs (one of the kinds of bugs included in PYBUGLAB) but either by randomly introducing the bugs in code or using a Cloze-like test. Instead, BUGLAB opts for a self-supervised approach and tackles a broader range of bugs. Concurrently to this work, Patra and Pradel [21] showed an alternative method for learning to generate realistic bugs. Dinella et al. [9] learn a supervised sequential model that performs graph transformations that replicate small edits in code (refactoring, introducing functionality, bug fixing, *etc.*). Their model — Hoppity — could serve as a learnable rewrite operation in BUGLAB in future work. Dynamic analysis methods have also been researched with promising results [31], but collecting representative dynamic traces over a diverse set of programs at scale (*e.g.* from the top Python packages used in this work) is practically impossible.

BUGLAB is related to ideas around self-supervised learning recently explored in deep learning, computer vision, and NLP. In our case, we aim to train a bug detection model without using training data from real-life bugs. BUGLAB resembles ELECTRA [6], with the important difference that the rewrites to the input code go beyond single token replacement that need to respect strict constraints of programming languages (syntax, variable scopes) and the model is directly used for bug detection, rather than for pre-training. The main BUGLAB objective Eq. 1 also resembles GANs [12] with the exception that the objective is non-differentiable (introducing a bug alters the discrete data representation), the selector is a structured probabilistic code rewriting model, and that we are mainly interested in the bug detector (analogous to the discriminator) rather than the selector.

## 7 Discussion and Conclusions

Learned program analyses offer the promise to improve how we develop software. They also offer a great opportunity to study machine learning models that combine formal and probabilistic reasoning. Towards achieving these we presented BUGLAB, a self-supervised approach for learning program analyses, that improves upon baseline methods and detects bugs in real-life code. We also empirically show the limitations of existing bug-detecting machine learning methods, which suffer from impractical false-positive rates. Importantly, we show the large gap of performance of existing methods on corpora of randomly inserted bugs — commonly used in prior work — and real-life bugs.

## Acknowledgements

We want to thank Sebastian Nowozin and Marwin Segler for helpful discussions, Marwin Segler for comments on a draft of this work, and the anonymous reviewers for useful questions and suggestions. Finally, we would like to thank the contributors to the following open-source tools used: PyTorch [20], PyDriller [27], MessagePack, LibCST, Jedi, Kubernetes, Helm.

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
