# A   Model Architectures

We implemented our models in PyTorch, using a shared codebase for GNN and GREAT models. The shared code covers both the input modules (embedding tokens into vectors) and the subnetworks used for bug localization and rewrite scoring.

In particular, the embedding of tokens uses a subtokenization strategy in which subtokens are embedded separately, and the token embedding is obtained by using max pooling over the subtoken embeddings. We have also experimented with alternative strategies (token-level embeddings and character-level embeddings followed by 1D-CNNs to obtain a single representation), but found subtokens to work best. We use a subtoken vocabulary of size 15000 and consider at most the first 6 subtokens of a token (dropping the remainder, if more exist), and an embedding dimension that matches the hidden dimension $d$ of the used GNN/GREAT architecture.

We discuss the details of the used GNN/GREAT models in the subsections below.

For localization, we use an architecture similar to pointer nets [19]. Let $\ell \in L$ be the potential locations for rewrites, and $\boldsymbol{r}_\ell \in \mathbb{R}^d$ their corresponding representations as computed by the GNN/GREAT, and $\boldsymbol{r}_{\texttt{NoBug}}$ the representation of the special $\texttt{NoBug}$ location. We first compute a "location query" as maximum over the projected representations of all considered locations, and then use a simple 2-layer MLP to compute per-location scores $s_\ell$:

$$q = \max\{\mathbf{W}_q \boldsymbol{r}_\ell \mid \ell \in L\}$$
$$s_\ell = \mathbf{W}_{mlp,2}\, \sigma(\mathbf{W}_{mlp,1}(\boldsymbol{r}_\ell \| q)).$$

Here, $\mathbf{W}_q \in \mathbb{R}^{d \times d}$ is a learnable projection and $\mathbf{W}_{mlp,2} \in \mathbb{R}^{1 \times d}, \mathbf{W}_{mlp,1} \in \mathbb{R}^{d \times d}$ are the learnable weights of our MLP. We can then model our distribution $p_{loc}$ from Sec. 3 by a softmax over these scores.

To model the distribution $p_{rew}$, we use the rewrite-scoring functions described in Sec. 4 followed by a softmax.

For all models, we use dropout between message passing/GREAT layers with rate 0.2, and train using the Adam optimizer with learning rate 1e-4 and a linear warm-up of 800 steps, additionally clipping gradient norms at 0.5. Bug selectors sample the distribution $S(\cdot)$ with an epsilon-greedy policy, with epsilon 0.02.

## A.1   GNN Architecture

Our GNN architecture follows a standard message-passing graph neural network [11], *i.e.* each message passing layer is defined as

$$\boldsymbol{h}_{v_i}^{(t+1)} = f^{(t)}\left(\boldsymbol{h}_{v_i}^{(t)}, \bigoplus_{\forall v_j : v_i \xrightarrow{k} v_j} \left(m^{(t)}\left(\boldsymbol{h}_{v_i}^{(t)}, k, \boldsymbol{h}_{v_j}^{(t)}\right)\right)\right).$$

Let $H^{(t)} = [\boldsymbol{h}_{v_0}^{(t)}, \cdots, \boldsymbol{h}_{v_{|V|}}^{(t)}]$ be a $|V| \times D$ matrix containing the output node states for all nodes $v_i \in V$ we can write the GNN computation as $H^{(t+1)} = \textsc{Gnn}(H^{(t)})$.

Our specific GNN message passing layers uses the structure defined as follows. Messages are computed as

$$m^t\left(\boldsymbol{h}_{v_i}^{(t)}, k, \boldsymbol{h}_{v_j}^{(t)}\right) = W_k^{(t)}\left[\boldsymbol{h}_{v_i}^{(t)}, \boldsymbol{h}_{v_j}^{(t)}\right], \tag{3}$$

*i.e.* a linear layer of the concatenation of the source and target node representations at time $t$ and $W_k^{(t)}$ is a edge type-specific linear layer. We use element-wise max pooling operator as $\bigoplus$. The node update function is defined as

$$f^{(t)}(\cdot) = \tanh\left(W_f^{(t)} \cdot \textsc{LayerNorm}\left(\textsc{Gelu}(\boldsymbol{m})\right) + \boldsymbol{b}_f\right),$$

where $W_f$ and $\boldsymbol{b}_f$ are learnable parameters and $\boldsymbol{m}$ is the output of the aggregation of the messages in Eq. 3 Backwards edge types are added for each existing edge type $k$, as in Li et al. [18].

Table 7: Accuracies on VarMisuse data of Hellendoorn et al. [13].

|  | Localization (on buggy data) | Localization (on non-buggy data) | Repair |
|---|---|---|---|
| GREAT-6L [13] | 86.14% | 88.98% | 85.85% |
| GREAT-6L (ours) | 86.10% | 93.33% | 89.69% |
| GREAT-10L [13] | 87.61% | 89.72% | 87.41% |
| GREAT-10L (ours) | 89.04% | 93.47% | 91.84% |

Table 8: Results of GREAT-based PYBUGLAB models in fully supervised setting.

|  | Joint | Loc | Repair |
|---|---|---|---|
| GREAT-5L | 48.63% | 59.30% | 75.66% |
| GREAT-6L | 46.73% | 57.68% | 74.90% |
| GREAT-7L | 43.33% | 54.08% | 73.20% |
| GREAT-8L | 51.04% | 61.87% | 76.26% |
| GREAT-9L | 47.91% | 58.84% | 75.10% |
| GREAT-10L | 47.86% | 58.67% | 75.28% |

We use 8 GNN layers like the one discussed above but with residual layers, *i.e.*

$$H^{(4)} = \text{GNN}_4 \left( \left[ H^{(0)}, \text{GNN}_3 \left( \text{GNN}_2 \left( \text{GNN}_1 \left( H^{(0)} \right) \right) \right) \right] \right)$$
$$H^{(8)} = \text{GNN}_8 \left( \left[ H^{(4)}, \text{GNN}_7 \left( \text{GNN}_6 \left( \text{GNN}_5 \left( H^{(4)} \right) \right) \right) \right] \right),$$

where the concatenations of the residual layers is over the node vectors. Finally, we set the representation of each entity $v_i$ as $\boldsymbol{r}_{v_i} = \boldsymbol{h}_{v_i}^{(8)}$.

We use a node hidden size of 256 and a minibatch size up to 300 graphs with no more than 10000 nodes in total.

## A.2 GREAT Architecture

We re-implemented GREAT [13] in PyTorch, following the paper and consulting the TensorFlow reference implementation where necessary. We verified that our implementation matches the reference implementation by using the model to train for the VarMisuse task defined for the dataset released by Hellendoorn et al. [13]. To this end, we considered two model configurations (6 and 10 layers, both with hidden representation size 512). The results are shown in Tbl. 7, indicating that our implementation matches (and in some regards, even outperforms) the reference implementation.

However, in our main PYBUGLAB experiments, we found that our GREAT models were usually outperformed by their GNN equivalents, contradicting earlier results by Hellendoorn et al. [13]. We tried to tune hyperparameters (such as number of layers) on the fully supervised bug detection detection dataset (row "Supervised" in Tbl. 1), first varying the number of layers from 5 to 10. This yielded the results shown in Tbl. 8. From the lack of a trend in these results we concluded that model capacity is not a limiting factor, and our reproduction results on the original GREAT data indicated that no implementation bugs in the GREAT layers needed to be suspected. As everything but the core code entity representation subnetwork is shared with the GNN models, which do not show such behavior, we ruled out implementation issues overall. Finally, we experimented with methods to stabilize training of deep Transformer networks such as ReZero [5] and LayerScale [29], and varying where LayerNorm is applied (before/after each sublayer). All of these experiments did not show significant improvement.

Consequently, our main results in Sec. 5 are reported on the GREAT configuration performing best in the fully supervised setting; yielding 8 layers with a hidden representation size of 512, 8 heads, and an intermediate size of 2048. During training, we have set the the maximum sequence length to 400 and used a minibatch size of 20.

Table 9: List of Entity (Node) Types in PYBUGLAB Representation.

| Entity Type | Description |
|---|---|
| Token | A token in the Python. |
| SyntaxNode | An AST node as defined in libCST's concrete syntax trees. |
| Type | The fully-qualified name of a type inferred by Jedi. |
| Documentation | The full docstring comment of an invoked method. |
| Symbol | A symbol (variable, function, *etc.*) in Python's symbol table. |
| Subtoken | A subtoken of an identifier, deterministically split on `camelCase` and `pascal_case`. |
| FormalArgName | The name of a formal argument of a method declaration. |

Table 10: List of Relationship (Edge) Types in PYBUGLAB Representation.

| Relation Type | Description |
|---|---|
| NextToken | Links two consecutive Token nodes. |
| SyntaxChild | Links a parent SyntaxNode to its child SyntaxNode or Token. |
| SyntaxNextSibling | Links a SyntaxNode to its subsequent sibling. |
| Type | Links a Symbol to its candidate type, if one has been inferred. |
| CallDoc | Links a method invocation SyntaxNode to its candidate Documentation. |
| FormalArg | Links an argument SyntaxNode to its FormalArgName. |
| ControlFlowNext | Link an statement SyntaxNode to a potentially succeeding statement SyntaxNode in terms of control flow. When branching occurs, a statement my have multiple links to other statements. |
| AssignedFrom | Links a target value SyntaxNode to the expression SyntaxNode syntax node. |
| ReturnsFrom | Links a function definition SyntaxNode to a return statement SyntaxNode it contains. |
| YieldsFrom | Links a generator definition SyntaxNode to a yield statement SyntaxNode it contains. |
| OccurenceOf | Links a variable Token or attribute SyntaxNode to the Symbol node it refers to. |
| LastMayUse | Links a usage of a variable Token or attribute SyntaxNode to all the potential immediately previous usages. |
| LastMayWrite | Links a usage of a variable Token or attribute SyntaxNode to all the last potential write operations. |
| MayFinalUseOf | Links any potential last usage of a variable Token or attribute SyntaxNode to its Symbol node. |

# B   Python Code Representation

To extract the entities and relationships we use libCST and Jedi that either directly provide the necessary data or allow to compute them. Tbl. 9 briefly describes the included entities and Tbl. 10 the relationships among the entities. Most of those entities and relationships are first used or inspired from Raychev et al. [25], Allamanis et al. [3, 4], Wei et al. [33], Cvitkovic et al. [7].

For the synthetic code snippet of Fig. 2, Fig. 4 shows the entities and relationship within that snippet. The signature of bar is set to `def bar(formal_bar_arg1, formal_bar_arg2)` for illustration purposes.

# C   PYBUGLAB Evaluation Metrics

Fig. 5 shows the definitions of the evaluation metrics used in this work.

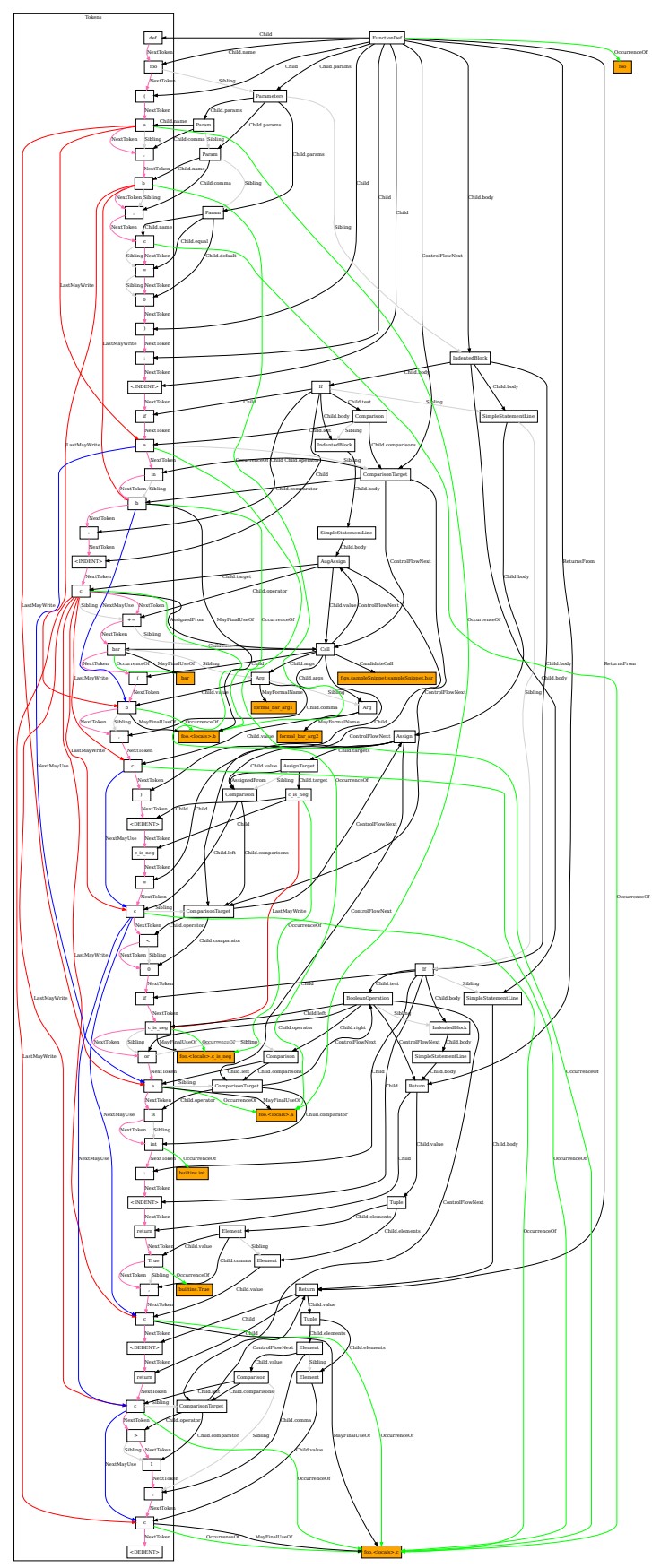

Figure 4: The graph representation of the entities and relationships for the snippet of Fig. 2. Please zoom this pdf.

| Detection False Warnings (DFW) | = | PredictedLoc $\neq$ GroundLoc AND PredictedLoc$\neq$NoBug |
|---|---|---|
| Detection True Warnings (DTW) | = | PredictedLoc $=$ GroundLoc AND GroundLoc$\neq$NoBug |
| False Detection Rate (FDR) | = | DFW / (DFW + DTW) |
| Detection Precision (DPr) | = | 1 - FDR |
| Detection Recall (DRe) | = | DTW / (# buggy samples) |
| Repair Accuracy Given Location (RAcc) | = | (# correct rewrite predictions) / (# buggy samples) |
| Detect and Repair True Warning (TW) | = | LTW AND PredictedRewrite = GroundRewrite |
| Detect and Repair False Warning (FW) | = | DFW OR (PredictedLoc$\neq$NoBug AND PredictedRewrite $\neq$ GroundRewrite) |
| Detect and Repair Precision (Pr) | = | TW / (TW + FW) |
| Detect and Repair Recall (Re) | = | TW / (# buggy samples) |

Figure 5: Evaluation Metrics for Bug Detection and Repair

# D PYPIBUGS Description

We collected a dataset, called PYPIBUGS, by crawling all commits across all PyPi python packages found in the libraries.io v1.6.0 dataset [17]. If any of the rewrites of Sec. 4.1 yields a change that is identical (modulo formatting and comments) to the observed commit, we consider it a candidate bug fix.

This yielded around 9000 changes. Note that here, to increase the precision of the dataset we require that a change is the sole code change within a file of code. This excludes cases where more than one bug-fixing change happened within a file, but also removes many changes that are not bug fixes.

The collected commits, along with the commit message, were then shown to one of the authors of this paper to filter out any non-bug fixing snippets. The annotation guideline was to look at the diff and the commit message and reject it if it is does *not* look like a bug-fixing commit. Through this process 2377 changes remain. The human annotation process removed various changes that mainly concerned configuration-like "constants", mainly literals such as changing the verbosity level, changing the default values of flags and arguments, turning off or on different parts of the software, *etc.* It also removed other deployment-related artifacts such as the version number of the software. Finally, we checked out each software project and attempted to extract our different code representations, which removed another 3 samples due to parsing issues. The result is PYPIBUGS, a highly sanitized dataset of 2374 samples representing a large variety of real-world "stupid simple bugs".

Note that PYPIBUGS differs from ManySStuBs [16] in many ways. First our dataset only includes Python code exposing the issues that can be represented by the different types of rewrites discussed in Sec. 4.1. It is also aims to achieve high precision, by looking at sole changes within a single file and commit and using manual annotation to filter changes that do not fix bugs. The breakdown of the different kinds of bugs in the dataset follows:

Table 11: Kinds of Bugs in PYPIBUGS

| Bug Kind | Num | Pct (%) |
|---|---|---|
| Argument Swapping | 283 | 11.9 |
| Wrong Assignment | 45 | 1.9 |
| Wrong Binary Operator | 81 | 3.4 |
| Wrong Boolean Operator | 192 | 8.1 |
| Wrong Comparison Operator | 407 | 17.1 |
| Wrong Literal | 88 | 3.7 |
| Variable Misuse | 1278 | 53.8 |
| Total | 2374 | (100%) |

**Replication of dataset**  Due to licensing concerns, we cannot release the raw data. Instead, we provide the GitHub git URLs of the projects along with the commit SHAs, filepaths, and bug types as a `jsonl` file. Anyone wishing to replicate the dataset can do so by cloning the projects and looking for the appropriate commits. We also provide a Python script in the supplementary material. The

script automates the whole cloning and checkout process, but requires the user to implement the `visit_buggy_code` and `visit_fixed_code` with code extracting the appropriate representation of the code for each of the bugs in PYPIBUGS.

## E   RANDOMBUGS Description

The following table contains the statistics per kind of bug for the the RANDOMBUGS dataset.

Table 12: Kinds of Bugs in RANDOMBUGS

| Bug Kind | Num | Pct (%) |
|---|---|---|
| Argument Swapping | 58459 | 7.7 |
| Wrong Assignment | 58821 | 7.7 |
| Wrong Binary Operator | 16848 | 2.2 |
| Wrong Boolean Operator | 15070 | 2.0 |
| Wrong Comparison Operator | 57037 | 7.5 |
| Wrong Literal | 80025 | 10.5 |
| Variable Misuse | 405950 | 53.3 |
| NOBUG | 69235 | 9.1 |
| Total | 761445 | (100%) |

## F   Additional Evaluation Results

Some additional evaluation results are included in this appendix.

### F.1   Localization & Repair Assuming Code is Buggy

In this subsection, we mask out the NOBUG option for both the GNN and GREAT models and ask these models to localize and repair bugs *only* in buggy code. The results are shown in Tbl. 13.

## G   Detected Bugs in Open-Source Projects

The following real-life bugs were detected by PYBUGLAB and a pull request was submitted.

### G.1   Bug in `spulec/moto`

```
@@ -45,7 +45,7 @@ def describe_identity_pool(self, identity_pool_id):
    identity_pool = self.identity_pools.get(identity_pool_id, None)
```

Table 13: Localization (%) per bug kind for the PYBUGLAB +Aug models when masking out the NOBUG option. This is similar to Tbl. 3 but only buggy examples are input to the models and they are *disallowed* to predict NOBUG. Note that repair results in Tbl. 3 are intact.

| Bug Type | RANDOMBUGS | | PYPIBUGS | |
|---|---|---|---|---|
| | GNN | GREAT | GNN | GREAT |
| Argument Swapping | **86.5** | 72.1 | **41.0** | 39.3 |
| Wrong Assign Op | 92.6 | **95.5** | **20.0** | 18.6 |
| Wrong Binary Op | **84.7** | 81.7 | 33.3 | **45.1** |
| Wrong Boolean Op | **74.3** | 49.3 | **32.8** | 24.7 |
| Wrong Comparison Op | **85.4** | 84.0 | 41.3 | **46.5** |
| Wrong Literal | **73.8** | 72.3 | **29.5** | 24.3 |
| Variable Misuse | **85.5** | 80.3 | 37.9 | **40.1** |
| **All Bug Kinds** | **84.9** | 79.6 | 37.6 | **39.0** |

```
        if not identity_pool:
-           raise ResourceNotFoundError(identity_pool)
+           raise ResourceNotFoundError(identity_pool_id)

        response = json.dumps(
            {
```

Pull request: `https://github.com/spulec/moto/pull/3582` (Merged)

### G.2   Bug in `apache/tinkerpop`

```
@@ -64,7 +64,7 @@ def __init__(self, partition_key=None, write_partition=None, read_partitions=Non
        self.configuration["partitionKey"] = partition_key
    if write_partition is not None:
        self.configuration["writePartition"] = write_partition
-   if write_partition is not None:
+   if read_partitions is not None:
        self.configuration["readPartitions"] = read_partitions
    if include_meta_properties is not None:
        self.configuration["includeMetaProperties"] = include_meta_properties
```

Pull request: `https://github.com/apache/tinkerpop/pull/1379` (Merged)

### G.3   Bug in `certbot/certbot`

```
@@ -166,7 +166,7 @@ def probe_sni(name, host, port=443, timeout=300, # pylint: disable=too-many-argu
    " from {0}:{1}".format(
        source_address[0],
        source_address[1]
-   ) if socket_kwargs else ""
+   ) if any(source_address) else ""
)
socket_tuple = (host, port)  # type: Tuple[str, int]
sock = socket.create_connection(socket_tuple, **socket_kwargs)  # type: ignore
```

Pull request: `https://github.com/certbot/certbot/pull/8605` (Merged)

Note that here PYBUGLAB detected a variable misuse bug, however the repair was more nuanced that replacing `socket_kwargs` with `source_address`.

### G.4   Bug in `Polyconseil/aioamqp`

```
@@ -305,7 +305,7 @@ def _close_channels(self, reply_code=None, reply_text=None, exception=None):
    if asyncio.iscoroutinefunction(self._on_error_callback):
        asyncio.ensure_future(self._on_error_callback(exception), loop=self._loop)
    else:
-       self._on_error_callback(exceptions.ChannelClosed(exception))
+       self._on_error_callback(exception)

for channel in self.channels.values():
    channel.connection_closed(reply_code, reply_text, exception)
```

Pull request: `https://github.com/Polyconseil/aioamqp/pull/224` (Open)

### G.5   Bug in `apache/beam`

```
@@ -636,7 +636,7 @@ def test_track_pcoll_unbounded(self):
    pcoll2 = pcoll1 | 'do1' >> FlatMap(lambda x: [x + 1])
    pcoll3 = pcoll2 | 'do2' >> FlatMap(lambda x: [x + 1])
    self.assertIs(pcoll1.is_bounded, False)
-   self.assertIs(pcoll1.is_bounded, False)
+   self.assertIs(pcoll2.is_bounded, False)
    self.assertIs(pcoll3.is_bounded, False)

  def test_track_pcoll_bounded(self):
```

Pull request: `https://github.com/apache/beam/pull/13761` (Merged)

## G.6 Bug in `sarugaku/requirementslib`

```
@@ -487,15 +487,13 @@ def get_dependencies_from_index(dep, sources=None, pip_options=None, wheel_cache
    session, finder = get_finder(sources=sources, pip_options=pip_options)
    dep.is_direct = True
    requirements = None
-   setup_requires = {}
    with temp_environ(), ExitStack() as stack:
        if not wheel_cache:
            wheel_cache = stack.enter_context(_get_wheel_cache())
        os.environ["PIP_EXISTS_ACTION"] = "i"
        if dep.editable and not dep.prepared and not dep.req:
            setup_info = SetupInfo.from_ireq(dep)
            results = setup_info.get_info()
-           setup_requires.update(results["setup_requires"])
            requirements = set(results["requires"].values())
        else:
            results = pip_shims.shims.resolve(dep)
```

Pull request: `https://github.com/sarugaku/requirementslib/pull/282` (Open)

Note that the bug detected by PYBUGLAB is caused by dead/unused code with this pull request removes.

## G.7 Bug in `CiscoDevNet/webexteamssdk`

```
@@ -233,7 +233,7 @@ def update(self, roomId, title, **request_parameters):
    """
    check_type(roomId, basestring)
-   check_type(roomId, basestring)
+   check_type(title, basestring)

    put_data = dict_from_items_with_values(
        request_parameters,
```

Pull request: `https://github.com/CiscoDevNet/webexteamssdk/pull/150` (Merged)

## G.8 Bug in `percolate/redset`

```
@@ -250,7 +250,7 @@ def _pop_items(self, num_items):
    try:
        res.append(self._load_item(item_str))
    except Exception:
-       log.exception("Could not deserialize '%s'" % res)
+       log.exception("Could not deserialize '%s'" % item_str)

return res
```

Pull request: `https://github.com/percolate/redset/pull/12` (Open)

## G.9 Bug in `pytorch/pytorch`

```
    if extra_inputs:
        extra_input_names, extra_input_sizes = zip( *extra_inputs)
-       extra_inputs = _RectifyNames(extra_input_names)
+       extra_input_names = _RectifyNames(extra_input_names)
        extra_inputs = zip(extra_input_names, extra_input_sizes)
```

Issue: `https://github.com/pytorch/pytorch/issues/51410` (Open, Triaged)

## G.10 Two Bugs in `saltstack/salt`

```
@@ -494,7 +494,7 @@ def enable( **kwargs):
        if "enabled" in beacons and beacons["enabled"]:
```

```
             ret["result"] = True
             ret["comment"] = "Enabled beacons on minion."
-        elif event_ret:
+        elif "enabled" in beacons and not beacons["enabled"]:
             ret["result"] = False
             ret["comment"] = "Failed to enable beacons on minion."
         else:
```

```
@@ -546,7 +546,7 @@ def disable( **kwargs):
             if "enabled" in beacons and not beacons["enabled"]:
                 ret["result"] = True
                 ret["comment"] = "Disabled beacons on minion."
-            elif event_ret:
+            elif "enabled" in beacons and beacons["enabled"]:
                 ret["result"] = False
                 ret["comment"] = "Failed to disable beacons on minion."
             else:
```

Pull request: https://github.com/saltstack/salt/pull/59381 (Open)

## G.11 Bug in mahmoud/botons

```
@@ -286,7 +286,7 @@ class DeferredValue(object):
    """
   def __init__(self, func, cache_value=True):
       self.func = func
-      self.cache_value = True
+      self.cache_value = cache_value
       self._value = _UNSET

   def get_value(self):
```

Pull request: https://github.com/mahmoud/boltons/pull/277 (Merged)

## G.12 Bug in geopy/geopy

```
@@ -426,8 +426,6 @@ def testContentAttrib(selector, key):
       for key, value in iter(el.items()):
           if value is not None:
               place[key] = value.text
-              if value.text is None:
-                  place[key] = None
           else:
               place[key] = None
```

Pull request: https://github.com/geopy/geopy/pull/469 (Merged)

## G.13 Bug in allure-framework/allure-python

```
@@ -51,7 +51,7 @@ def parse_tag(tag, issue_pattern=None, link_pattern=None):
       if issue_pattern and kind == "issue" and not value.startswith("http"):
           value = issue_pattern.format(value)
       if link_pattern and kind == "link" and not value.startswith("http"):
-          value = issue_pattern.format(value)
+          value = link_pattern.format(value)
       return Link(type=kind, name=name or value, url=value)

   if __is(kind, LabelType):
```

Fixed in-between: https://github.com/allure-framework/allure-python/commit/
34a91fa1f32e9f5279f14a595cb5401469b75ad8

## G.14 Bug in qiniu/python-sdk

```
@@ -38,7 +38,7 @@ def set_default(
```

```
        if default_api_host:
            _config['default_api_host'] = default_api_host
        if default_uc_host:
-           _config['default_uc_host'] = default_api_host
+           _config['default_uc_host'] = default_uc_host
        if connection_retries:
            _config['connection_retries'] = connection_retries
        if connection_pool:
```

Fixed in-between: https://github.com/qiniu/python-sdk/commit/ 71cf09cc04060524b4835a9b5d45a8ae3a4483c6