# OpenReview forum: "Self-Supervised Bug Detection and Repair"
_NeurIPS.cc/2021/Conference — NeurIPS 2021 Poster_

### Official Review · Reviewer_e6hN · 2021-07-09

**Rating:** 7
**Confidence:** 4

**Summary:**

This work tackles programming bug detection and repair, introducing BugLab, which uses two models trained in an adversarial fashion together. One model is a bug detector that learns to identify and repair bugs, while the other is a bug selector that introduces buggy mutations in code that are difficult for the detector to identify. When trained together, the detector model is able to see more training data for bugs that it is worse at detecting, leading to better performance after sufficient training in this fashion. BugLab also uses semantics-preserving code modifications as a form of data augmentation. The experiments show that the BugLab training procedure leads to improved performance in bug localization and repair when applied to prior models (GNN and GREAT), compared to baselines of supervised learning and selecting buggy mutations at random instead of sampling from the trained selector model.

**Limitations And Societal Impact:**

The authors did a good job here.

**Main Review:**

## Positives:
* Intuitive idea of training two models together such that they help each other train better, which is easily understood because it is similar to other works like GANs
* The main idea is presented clearly and well-justified
* Detailed analysis of experiments

## Negatives:
* The experiments have some minor gaps, although I don’t expect these gaps to affect the conclusions drawn by the paper
* The text could use some editing for clarity and typos

---

**Originality: 3/5** The idea of using two models trained together is not new (nor did the authors claim that this is new), but this is the first work I’m aware of that applies this technique to code repair. I think the technique is particularly applicable here specifically due to the lack of large datasets of real bugs, and random and synthetically generated bugs likely leads to a distribution quite different from real bugs. By using a bug selector to choose bugs that the detector struggles with, the detector is trained on increasingly difficult bugs and has more of its bases covered.

**Quality: 4/5** The paper is generally well written and the experiments and analysis are generally good too. However there are some gaps (errors in experiment setup or obvious questions left unanswered) in the experiments that prevent them from feeling complete, although I suspect the main conclusions of the paper to remain the same regardless. See questions below.

**Clarity: 3/5** The paper communicates its main ideas clearly, but some details could use clarification or better wording (see below). The paper also introduces a lot of notation without many concrete examples to ground the notation with ideas in the reader’s mind, and I am wondering if all of the notation is actually necessary, or if some parts of the presentation could be simplified and made more intuitive.

**Significance: 4/5** Code repair is a significant research topic with obvious practical impact. Although BugLab is not yet good enough for use in real coding environments due to its impractical false positive rate, it makes good progress toward a significant goal.


## Questions and comments
The most important items, in my opinion, are marked with **(!)**.
* Line 181: do you consider swapping `a == b` with `b == a`? What about swapping operands of commutative operations like + or * (on numbers, not on sequences)?
* **(!)** Line 196-205: What do the “pools” of training data actually contain? What specifically are the “new elements for the data pools”? Are there separate data pools for training D versus S?  I read this paragraph a couple times but didn’t quite understand the details, so clarification here would be helpful.
* **(!!)** 294: CuBERT considers more “operators” in its WrongOp task than just l11 and l20. In particular, `in`, `not in`, `is`, `is not`, `and`, `or`, etc. are all “operators” considered by CuBERT, and thus the bug locations could be l2, l22, l14, l16, and l20 -- see Table 7 of the CuBERT paper for a more complete list. The corresponding experiment should be updated. _I view this as high priority to fix, since otherwise the paper would be making factually incorrect claims about CuBERT._ Adequately addressing this issue will improve my opinion of the paper -- I think this is a good paper overall but, at the moment, I can’t suggest acceptance due to the factually incorrect claims.
* **(!)** Why no NoBug category for PyPIBugs? Considering that PyPIBugs consists of code that was updated with a rewrite captured by Sec. 4.1, and we’re assuming that the “before” version of the code is buggy, I think it’s similarly safe to assume that the “after” version of the code is bug-free.
* Section 4.1: Similar to Argument Swapping, why not consider Operand Swapping? I.e., instead of changing the order of arguments to a function call, consider swapping the operands on the two sides of a binary operator. (This is the Swapped Operand task in CuBERT.)
* Checklist 4(b): Did you check that all of the individual open-source licenses are compatible with how you’re using+releasing the code? For example, viral licenses may require that your dataset/code also be released under the same viral license, which may cause conflicts with requirements imposed by other licenses.

## Minor Suggestions
* Line 93: The text says relations in E are mapped to “elements of V_tok” (i.e., individual tokens) but the elaboration seems to imply that each relation is an edge between 2 elements of V_tok
* Line 116: When looking for “three concrete rule score functions in Sec. 4”, I wanted to find mention of $w_p$ since this is the notation for rule score functions (line 113). Although I believe I know what the 3 rule score functions are in Sec. 4, this $w_p$ notation is not used.
* Line 209: On a first read, I thought, “Why are some packages excluded from training?” The question is answered on line 221 (the remaining packages form the test set), but I think clarification could be added to line 209
* 259: double quotes going the wrong way
* 270: “results the” -> “results of the”
* 274: “seems to is” -> “seems to be”
* 306: the truncation is in line 4
* 378: “offer” -> “offers”, since “analysis” is singular
* 383: “relative” -> “relatively”

---
Update after author response: changed score from 5 to 7

**Time Spent Reviewing:**

3.5

---

> ### Author Response · Authors · 2021-08-10
> **Author Response**
>
> Thank you for your time and your feedback for our submission. Here, we try to address the main points raised in your review (we will simply integrate the minor suggestions in the next revision)
>
> > Line 181: do you consider swapping a == b with b == a? What about swapping operands of
> > commutative operations like + or * (on numbers, not on sequences)?
>
> We do swap a == b and a != b (and also mirror a > b > c into c < b < a), but do not do
> this for binary operations.
>
> > Line 196-205: What do the “pools” of training data actually contain?
>
> They are equivalent to $C^{(i)}_D$ / $C^(i)_S$ in Alg. 1 (i.e. contain pairs of code and labels).
>
> > What specifically are the “new elements for the data pools”?
>
> From time to time, we take a snapshot of the current models S/D and generate new samples (as
> in lines 3/6 of Alg. 1) in separate workers, while we continue to run our training. The
> new samples are then added to the pool.
>
> > Are there separate data pools for training D versus S?
>
> Yes.
>
> > I read this paragraph a couple times but didn’t quite understand the details, so clarification
> > here would be helpful.
>
> We will try to improve the wording of this paragraph after the discussion with you, to
> make sure that we cover all potential misunderstandings.
>
> > 294: CuBERT considers more “operators” in its WrongOp task than just l11 and l20. In
> > particular, in, not in, is, is not, and, or, etc. are all “operators” considered by CuBERT,
> > and thus the bug locations could be l2, l22, l14, l16, and l20 -- see Table 7 of the CuBERT
> > paper for a more complete list. The corresponding experiment should be updated.
>
> Thank you for catching this - this is entirely a bug in the writing. Just to clarify, the
> masking of allowed rewrites described in line 290 uses the following list of operators:
> ```python
> CUBERT_ALLOWED_OPERATORS = {
>     "+",
>     "*",
>     "-",
>     "/",
>     "%",
>     "==",
>     "!=",
>     "is",
>     "is not",
>     "<",
>     "<=",
>     ">",
>     ">=",
>     "in",
>     "not in",
>     "and",
>     "or",
> }
> ```
> Hence, we believe that the experimental results remain valid as they are, but we will
> fix the example discussion in the paper as you noted.
>
> > (!) Why no NoBug category for PyPIBugs? Considering that PyPIBugs consists of code that was
> > updated with a rewrite captured by Sec. 4.1, and we’re assuming that the “before” version of
> > the code is buggy, I think it’s similarly safe to assume that the “after” version of the code
> > is bug-free.
>
> This is a good idea! It still has some problems in that buggy samples would still be massively
> overrepresented in this dataset compared to the real world, but it would provide at least some
> more realistic quantitative evaluation of this aspect.
>
> We have now compared the models used in the evaluation of the paper on this restricted dataset. As this focuses the evaluation on the ability to classify samples correctly as buggy/non-buggy (and we cannot provide plots in this rebuttal), we are reporting the area under the precision/recall curve for all of our considered models, evaluated on the union of “PiPyBugs” (as in the paper) and “PiPyBugs-AfterFix” (as suggested by you):
>
> | Model | AUPRC |
> | -------- | ---------- |
> | GNN - Supervised | 0.0867 |
> | GNN - Random Selector | 0.1083 |
> | GNN - PyBugLab | 0.1595 |
> | GNN - PyBugLab+SemPr | 0.1870 |
> | GREAT - Supervised | 0.044 |
> | GREAT - Random Selector | 0.1171 |
> | GREAT - PyBugLab |  0.1398 |
> | GREAT - PyBugLab+SemPr | 0.1294 |
>
> > Section 4.1: Similar to Argument Swapping, why not consider Operand Swapping? I.e., instead
> > of changing the order of arguments to a function call, consider swapping the operands on
> > the two sides of a binary operator. (This is the Swapped Operand task in CuBERT.)
>
> We considered this, but it is not always clear when these are appropriate for many operators (e.g., “+” and “*” are commutative on scalar, but used for many other things as well). The required analysis for these cases would be substantial. This could be done more easily when restricting oneself to “-“, “/” and “%”, but we have decided (so far) that we believe it is unlikely that this would change our overall results. Of course, if we were aiming to implement a realistic bug finding tool, these would make sense, but for that, we first need to overcome the huge false positive rate.
>
> > Checklist 4(b): Did you check that all of the individual open-source licenses are compatible
> > with how you’re using+releasing the code? For example, viral licenses may require that your
> > dataset/code also be released under the same viral license, which may cause conflicts with
> > requirements imposed by other licenses.
>
> We do not plan to release the code (for reasons related to figuring out the permissions
> around redistribution); rather, we are releasing the dataset as a list of metadata
> that can be consumed by a script to download the data anew from PyPi, giving authors the
> opportunity to opt out by removing their code from PyPi.
>
> For using the code for training, we are relying on the usual ``fair use'' exception, but we
> are aware of the recent discussions around this w.r.t. OpenAI Codex / GitHub Co-Pilot.

---

> > ### Comment · Reviewer_e6hN · 2021-08-26
> > **Thanks for clarifications**
> >
> > I'm quite satisfied with these clarifications, provided they are reflected in the final paper.
> >
> > For the paragraph on lines 196-205, it makes sense to me now. My main confusion was getting tripped up by the term "pools" which was not well explained. I think all of the references to Alg. 1 in your clarification would be very helpful to add to the paper.
> >
> > I'm glad the CuBERT issue was merely a writing issue, without needing to redo the experiments completely.
> >
> > The new results for the PyPI dataset also help to strengthen the paper. I imagine this would be added as a new table in the paper, but don't forget to update Table 2 as well with NoBug numbers.
> >
> > Thanks for the detailed response! I'm increasing my score from 5 to 7.

---

### Official Review · Reviewer_vPhi · 2021-07-15

**Rating:** 9
**Confidence:** 5

**Summary:**

This paper describes an adversarial approach towards detecting and repairing local bugs in human-written software. The approach involves training in tandem two neural networks, the detector, which discovers and repairs bugs in a given piece of code, and the selector, which produces bugs that the detector is unable to detect. At test time, only the detector is used.

The architecture used for each of the networks is similar: first the program is represented by a graph with labeled edges, then each node is embedded using a language-like model (to take into account semantic information in variable names), then embeds the graph using either a GREAT relational transformer or a GNN. The code rewriting model then predicts a distribution over locations to rewrite using a pointer network, and a distribution of rewrites at that given location using a softmax over potential rewrites.

The types of bugs considered are local and shallow bugs such as variable misnamings or use of the wrong operator or literal in an expression. The authors create a corpus of code by crawling pypi and create a corpus of known shallow bugs by looking for version updates that include edits that fit their definition of a shallow bug. They additionally include a corpus of random generated bugs. In general, their approach works better on the random bug corpus than on the real bug corpus, and also works much better at repairing bugs given localization than it does at localization. They are able to achieve better performance than the special purpose CuBERT model using a F1 metric. Additionally, they were able to detect 19 bugs in open source projects, but with a TPR of 0.19% this is probably not yet a productionizable tool in this context.

**Limitations And Societal Impact:**

The paper very clearly addresses its limitations. I do not think the approach provided here has a substantial potential negative social impact (software tooling probably will not lead to job displacement or surveillance), so it’s probably fine that it is not addressed in detail.


**Main Review:**

Originality: the central topic of the paper, the adversarial approach, is novel in this domain as far as I can tell. The neural architectures are not original, but they are not the main focus of the paper and are adequately cited.

Quality: The paper makes claims that are clearly supported by the evidence provided in the paper, and does not shy away from presenting limitations. I especially appreciated the real world experimentation as well as  the clean structure of the training process and the similar neural architectures.

I have a few minor nits that are listed below:

121: I think mentioning earlier (perhaps towards the beginning of the introduction) that the bug set you are targeting is simple bugs might be helpful, as it contextualizes the local techniques being used. I would also avoid characterizing the bugs as “stupid simple bugs” and instead characterize them as local, as many bugs that are local are not simple to detect and many bugs that are simple to detect are not local.

296: If possible, a comparison to a CuBERT model that has been specifically conditioned to have the same precision as the PyBugLab model or vice versa would be slightly more convincing (just change one of the detection thresholds so that the precisions match). Not a huge deal if this is too difficult to do since there’s such a big gap in the recall.

Clarity: The submission is clearly written and conveys all the necessary information. In general it is easy to read and the key results being highlighted with the bug icon and underline makes it easier to quickly find them.

Significance: The results seem pretty significant. While I don’t know how much this specific task actually needs to be solved (as the authors themselves acknowledge), it certainly is a real world task. More importantly, the broader approach here is not at all specific to the kinds of bugs this paper tries to address. I can see this approach applied to bug repair of human programs and even potentially to domains such as program synthesis.


**Time Spent Reviewing:**

1.5

---

> ### Author Response · Authors · 2021-08-10
> **Author Response**
>
> Thank you for your review and your kind comments. Here, we try to address the main points you raised in your review.
>
> > 121: I think mentioning earlier (perhaps towards the beginning of the introduction) that the bug
> > set you are targeting is simple bugs might be helpful, as it contextualizes the local techniques
> > being used.
>
> This is a good point, and we will update the introduction to make this point.
>
> > I would also avoid characterizing the bugs as “stupid simple bugs” and instead characterize them
> > as local, as many bugs that are local are not simple to detect and many bugs that are simple to
> > detect are not local.
>
> "Simple Stupid Bugs" was coined by our reference [16], though we are happy to change that
> terminology. "Local" is maybe not a great choice either, as variable misuse and function argument
> swapping bugs require substantial non-local reasoning (about potentially distant variable uses
> and function definitions). The common feature here is that all of the considered bugs have a
> simple fix that can be performed using a single rewrite, but that doesn't lead to a pithy name.
> We are happy to take suggestions here, though :-)
>
> > 296: If possible, a comparison to a CuBERT model that has been specifically conditioned to have
> > the same precision as the PyBugLab model or vice versa would be slightly more convincing (just
> > change one of the detection thresholds so that the precisions match). Not a huge deal if this is
> > too difficult to do since there’s such a big gap in the recall.
>
> This is a good idea, and we will run an experiment like this over the next few days and provide these numbers.

---

> > ### Author Response · Authors · 2021-08-13
> > **CuBERT results update**
> >
> > After further communication with the CuBERT authors, we have refined our run script for their model slightly; this has improved results on variable misuse and worsened them on the wrong operator task. Concretely, table 3 should now read like this:
> >
> > | Dataset            | CuBERT - Prec | CuBERT - Recall | CuBERT - F1 | PyBugLab (GNN) - Prec | PyBugLab (GNN) - Recall | PyBugLab (GNN) - F1 |
> > | ------------------ | ------------- | --------------- | ----------- | --------------------- | ----------------------- | ------------------- |
> > | PyPiBugs-WrongOp   |     **0.764** |           0.251 |       0.378 |                 0.730 |               **0.764** |           **0.746** |
> > | PyPiBugs-Varmisuse |        0.632  |           0.403 |       0.493 |             **0.740** |               **0.840** |           **0.787** |
> >
> > If we calibrate the threshold for CuBERT such that recall matches the PyBugLab model, we obtain the following results:
> >
> > | Dataset            | CuBERT - Prec | CuBERT - Recall | PyBugLab (GNN) - Prec | PyBugLab (GNN) - Recall |
> > | ------------------ | ------------- | --------------- | --------------------- | ----------------------- |
> > | PyPiBugs-WrongOp   |        0.609 |            0.762 |             **0.730** |                   0.764 |
> > | PyPiBugs-Varmisuse |        0.613 |            0.840 |             **0.740** |                   0.840 |

---

### Official Review · Reviewer_aZJ8 · 2021-07-16

**Rating:** 5
**Confidence:** 4

**Summary:**

This paper proposes a self-supervised approach that trains bug detectors by co-training a bug selector that learns to create bugs. A high-level framework is formulated, and then a python implementation using graph neural networks and relational transformers is provided. Experiments on a manually curated bug dataset demonstrate its effectiveness.

**Limitations And Societal Impact:**

This paper has no potential negative societal impact.

**Main Review:**

The paper is well written, and the formula-based explanation of the method is clear.  It is an interesting idea of incorporating a bug selector with bug detectors, which resembles GAN though the overall framework is non-differentiable. But the overall training procedure seems very complicated. As mentioned by the authors, it costs more than one week to train the models.

The main problem of this paper is in the evaluation, making the experimental result somewhat unconvincing. Some important baselines (e.g., HOPPITY[1],  deepdebug[2]) are missing, making it difficult to verify the superiority of BUGLAB over existing techniques. Simply comparing with CuBert on only bug prediction task is somewhat weak. Or the authors can instantiate the BUGLAB framework based on other program-repair datasets used by previous works to verify the its generalization ability. I believe there can be a setting where BUGLAB can be compared with previous representative works, though it may involve tedious implementation work. Given current experimental results, it is not ready to be published at a high-profile conference.

-----After seeing the authors' response-----

In my view, though the method is claimed to be "self-supervised," the main contribution is more about a refinement strategy on auto-generated training data. Based on some heuristic rules, the bug selector will actually generate and select buggy code (of specific types ) that serves as training data for the bug detector. Some other works (e.g., DeepBugs) will also automatically generate training data (though not in a learning way), targeting similar restricted bug types (e.g., wrong operation and argument swapping) and based on similar rewrite rules. More comparison and analysis are necessary.  The data refinement idea does make sense, though its technical design is not novel to me.

[1] Elizabeth Dinella, Hanjun Dai, Ziyang Li, Mayur Naik, Le Song, and Ke Wang. Hoppity: Learning graph transformations to detect and fix bugs in programs. In International Conference on Learning Representations (ICLR), 2020.

[2] Dawn Drain, Chen Wu, Alexey Svyatkovskiy, and Neel Sundaresan. Generating bug-fixes using pretrained transformers. arXiv preprint arXiv:2104.07896, 2021. 4

**Time Spent Reviewing:**

4

---

> ### Author Response · Authors · 2021-08-10
> **Author Response**
>
> Thank you for your review, and your appreciation of the clarity of our presentation. Here, we answer the main points you raised.
>
> > Some important baselines (e.g., HOPPITY[1], deepdebug[2]) are missing, making it difficult
> > to verify the superiority of BUGLAB over existing techniques.
>
> We believe there is a slight misunderstanding here: the core contribution of our submission
> is a framework to train program repair models, not a new program repair model. Indeed, the
> models we used are kept intentionally simple while being representative; the intention is to
> show the benefits of our training methodology independent of the used model.
> We are not claiming that our models are superior to others, but that our training method helps
> to improve their performance compared to a simpler training setup (and compared to the
> pretrain-then-finetune approach of cuBERT).
>
> As we note in Sect. 6, more complex models such as Hoppity or DeepDebug could be trained
> in the BugLab framework as well, and we would expect that this would further improve their
> performance.
>
> > Or the authors can instantiate the BUGLAB framework based on other program-repair datasets
> > used by previous works to verify the its generalization ability.
>
> The BugLab framework is focused on the class of "stupid simple bugs", which makes it hard to
> integrate evaluations on the automatically, much less restricted datasets used in prior work
> (e.g., the JavaScript dataset used to evaluate Hoppity, or the Java dataset by Tufano et al.
> used to evaluate DeepDebug). [There's also the substantial overhead of supporting a different
> programming language.]
>
> However, we want to argue that PyPiBugs is a much better benchmark than these earlier datasets,
> as it is _manually_ curated rather than heuristically filtered. As we discuss in Appendix D,
> even after strict heuristic filtering of changes to the class of rewrites we wanted to consider,
> manual filtering removed a further 70% as changes that were not related to bugs.
> Hence, we believe that these existing datasets are largely unrepresentative of real-world bugs,
> and results on them are not indicative of real-world performance.

---

### Decision · Program_Chairs · 2021-09-28

**Decision:**

Accept (Poster)

**Comment:**

Using modern (deep unsupervised) methods to support bug detection in programs is an important direction. The reviewers generally liked the approach of this paper, though the relation of this method (for buggy data generation) to the methods actually used for bug detection means that proper evaluation is a bit subtle. Overall the final feeling was that this was sufficiently demonstrated to make the interesting techniques worth publishing. I would urge the authors though to pay careful attention to the points made by reviewer aZJ8 in revising the paper.

**Consistency Experiment:**

NeurIPS has a long history of experimentation. In 2014, NeurIPS ran an experiment in which 10% of submissions were reviewed by two independent committees to quantify the randomness in the review process. This year, we repeated a variant of this experiment to see how the quality of the review process has changed over time.  This paper was part of the experiment and was therefore assigned to two committees (consisting of reviewers, an Area Chair, and a Senior Area Chair) that reached independent decisions.  If both committees made the same recommendation, this recommendation was followed. If a single committee recommended acceptance, the paper was accepted (with the exception of a few cases in which the other committee identified what we considered a fatal flaw, e.g., an error in a key result).

Both committees reached the same decision: **Accept (Poster)**

The other committee assigned to the paper recommended **Accept (Poster)**.  You can find the other set of reviews, along with any follow up discussion with the authors here:
https://openreview.net/forum?id=zOngaSKrElL